

# Investigation of the relationships between peri-implant diseases, periodontal diseases, and conditions: a cross-sectional study

Tuğba Şahin

Division of Periodontology, Faculty of Dentistry, Abant Izzet Baysal University, Bolu, Turkey

## ABSTRACT

**Introduction:** Peri-implant and periodontal conditions share common underlying factors, including risk factors, microbiology, immunology, and treatment approaches.

**Aims:** This study aims to investigate the potential co-occurrence of peri-implant and periodontal conditions.

**Design:** One hundred twenty-three implants were divided into three groups: peri-implantitis (41 implants), peri-implant mucositis (41 implants), and peri-implant health (41 implants). Peri-implant and periodontal statuses were assessed using the 2017 AAP/EFP World Workshop on Classification of Periodontal and Peri-implant Diseases and Conditions. All measurements were performed by a single clinician (T.Ş.). One-way analysis of variance was used to compare the study groups according to the data. An assessment was conducted regarding the coexistence of periodontal and peri-implant conditions.

**Results:** Patients with peri-implant mucositis predominantly had gingivitis, whereas those with peri-implant health exhibited periodontal health. In contrast, patients with peri-implantitis mostly had gingivitis, with a lower occurrence of periodontitis. A significant difference was observed between the peri-implant and periodontal groups ($p = 0.003$). Significant differences were observed between peri-implant and periodontal evaluations for plaque indices, gingival indices, probing depth, gingival recession, and clinical attachment level ($p = 0.001$), ($p = 0.006$).

**Conclusions:** The findings of this study underscore the intricate influence of implant treatment on periodontal health. This observation emphasizes the importance of elucidating the underlying factors to improve clinical management and outcomes in patients with periodontal and peri-implant diseases, highlighting the relevance and potential impact of this research in the field.

## INTRODUCTION

Establishing and maintaining oral health involves monitoring both peri-implant and periodontal conditions. Periodontal health and peri-implant health are defined by the absence of bleeding, swelling, or suppuration on probing, coupled with the absence of clinically evident inflammation. In peri-implant mucositis, akin to gingivitis in natural dentition, the initiation of inflammation occurs with microbial plaque accumulation;

Corresponding author
Tuğba Şahin,
sahintugba1432@gmail.com

however, there is no extension of increased probing depth to the alveolar bone (*Berglundh et al., 2018*; *Chapple et al., 2018*). Peri-implant mucositis is distinguished from peri-implantitis by triggering a localized inflammatory response and ultimately leading to the loss of supporting bone around the implant (*Berglundh et al., 2018*; *Renvert et al., 2018*; *Lee & Wang, 2010*). Similarly, periodontitis, an enduring inflammatory condition, is typified by dysbiotic plaque biofilms and immune-dysregulation that promotes the destruction of the periodontal ligament and alveolar bone (*Sedghi, Bacino & Kapila, 2021*; *Papapanou et al., 2018*).

Periodontitis and peri-implantitis are inflammatory conditions caused by biofilms that can result in tooth and oral implant loss if not addressed (*Lasserre, Brecx & Toma, 2018*). Compared with healthy areas in the same individual, inflamed regions (peri-implantitis and periodontitis) harbor unique dysbiotic subgingival microbial ecosystems (*Barbagallo et al., 2022*). Periodontitis and peri-implantitis are reportedly associated with a notable increase in microbial stability within the subgingival microbiome (*Zhang et al., 2021*). Studies investigating peri-implant biofilms have focused predominantly on recognized periodontal pathogens such as *Porphyromonas gingivalis* (*P. gingivalis*) and *Treponema denticola*. Furthermore, these results highlight the similarities between the subgingival microbiota of periodontitis patients and those of peri-implantitis patients (*Kotsakis & Olmedo, 2021*). In contrast, certain studies refute the notion of microbiota similarity between peri-implantitis and periodontitis (*Koyanagi et al., 2013*; *Maruyama et al., 2014*). In a broad sense, risk factors encompass patient-related, environmental, or practitioner-related elements. Patient-related risk factors include socio-economic status, smoking habits, substance abuse disorders, diabetes, dietary habits and supplementation, mental health conditions, advanced age, inadequate home dental care, limited understanding of the importance of proper oral hygiene, genetic polymorphisms, and medication usage (*Darby, 2022*; *Kinane & Hart, 2003*; *Vaz et al., 2012*). Moreover, according to the report of the 6th Conference European Association for Osseointegration, prosthesis overcontouring and implant surface characteristics increase the risk of peri-implantitis (*Schwarz et al., 2021*).

Individuals with a history of periodontitis are more susceptible to peri-implant infections and complications (*Renvert & Persson, 2009*; *Ferreira et al., 2018*). A history of periodontitis can be assessed by evaluating periodontal bone loss on radiographs, examining dental records, or talking to the patient to determine the cause of tooth loss. It is reasonable to include the stage and extent of periodontal disease in this assessment as it influences the development and progression of peri-implant disease (*Heitz-Mayfield, Heitz & Lang, 2020*).

The peri-implant sulcus is histologically and immunologically distinct from the subgingival sulcus (*Robitaille et al., 2016*). Increasing evidence has been obtained on the development and causes of periodontal and peri-implant disorders. Although there are some similarities in the host's reactions in both settings, their differences can be attributed to the distinct compositions of tooth-periodontium and implant-alveolar bone biointerfaces (*Larsson et al., 2022*). The host response, which is pivotal in delineating the genetic basis of diseases like periodontitis and peri-implantitis, necessitates an examination

of cytokines, chemokines, growth factors, and their receptors, which is crucial in understanding the pathogenesis of periodontal and peri-implant diseases (*Turkmen & Firatli, 2022*; *Genco, 1992*).

Periodontal and peri-implant diseases are managed mainly *via* manual instrumentation to reduce the bacterial load and improve the patient's at-home cleanliness (*Meffert, 1996*). If needed, further antibiotic therapies and laser treatments may be employed (*Mombelli & Lang, 1992*; *Hammami & Nasri, 2021*; *Diwan et al., 2024*; *Ashnagar et al., 2014*). Regenerative treatment can also be applied (*Larsson et al., 2016*). Periodontitis and peri-implantitis reportedly have similar etiologies and similar therapeutic interventions are performed in patients with these two entities (*Robitaille et al., 2016*).

In particular, frequent co-occurrence of periodontitis and peri-implantitis, the manifestation of gingivitis alongside peri-implant mucositis, and the observation of periodontal health alongside peri-implant health are anticipated. There is no comparison in the literature regarding the simultaneous occurrence of all periodontal and peri-implant conditions. The objective of this study is to investigate the potential relationships between periodontal and peri-implant conditions. Given the commonalities in primary factors, risk factors, microbiology, immunology, and treatment interventions, this study hypothesizes that there is a significant association between the presence of periodontitis and peri-implantitis, between gingivitis and peri-implant mucositis, and between periodontal health and peri-implant health. Specifically, this study aims to determine whether periodontitis is more prevalent among patients with peri-implantitis, gingivitis is more common among those with peri-implant mucositis, and periodontal health is associated with peri-implant health.

# MATERIALS AND METHODS

## Ethical statement

Participants were recruited from the patients who were previously examined for their periodontal or peri-implant status at Bolu Abant İzzet Baysal University between July 2022 and July 2023. Patient records were reviewed to identify potential participants who met the inclusion criteria. Patients who agreed to participate were contacted, and informed consent was obtained before any further assessments were conducted. The Bolu Abant İzzet Baysal University Clinical Research Ethics Committee approved the study (2022/163). The participants were informed about the procedures and signed an informed consent form. Compliance with the STROBE guidelines for cross-sectional studies was documented. This clinical study is registered retrospectively at ClinicalTrials.gov (NCT06128850/ 31.03.2023). This study complied with the World Medical Association Declaration of Helsinki for medical research (*Emanuel, 2013*).

## Study design

A total of 167 patients were initially screened on the basis of their medical and dental records. The inclusion and exclusion criteria were applied during an initial screening of medical and dental records. The study included 123 implants in 69 patients with fixed prostheses who had survived for at least one year following functional prosthetic loading,
with the exception of patients with uncontrolled medical issues and referred clinical bruxism. The implants were 2–5 years old and the fixed prosthesis of the implant was 1.5–4.5 years old. Two hundred twenty-four patients were evaluated, and 123 implants from in sixty-nine people who met the inclusion criteria were included in the study.

## Inclusion and exclusion criteria

This study included systemically healthy patients aged 18–70 years who had undergone at least one year and at most five years of functional prosthetic loading of one or more dental implants with a fixed prosthesis. Pregnant or lactating women, patients with a history of chronic use of anti-inflammatory agents, and those on immunosuppressive drugs or drugs that impact the mucosa and bones were not included in the study. Patients who underwent treatment for peri-implant disease after implant placement, those with residual cement residue and prosthesis design, and those with malpositioned implants were also excluded from the study. Patients who underwent active periodontal treatment or treatment after implantation, diabetic patients, patients with mucosal diseases, and smokers were excluded.

## Sample size calculation

Three groups were planned according to the peri-implant health status in the study, and the sample size was calculated according to the methods of *Barwacz et al. (2018)*. According to the results of the power calculation *via* the F test, fixed effects, special effects, main effects, and interaction analysis (G * Power 3.1 software; Heinrich Heine University, Dusseldorf, Germany), with $\alpha$ (margin of error) = 0.05, power $(1 - \beta) = 0.90$ and effect size (f) = 0.4, the required sample size for the three groups was 123, and the required sample size for each subgroup was at least 41. The effect size value was determined according to the proposed large effect size convention.

## Data collection and measurements

The plaque index (*Silness & Löe, 1964*), gingival index (*Loe & Silness, 1963*), probing depth, bleeding on probing (*Ainamo & Bay, 1975*), clinical attachment loss (CAL), and gingival recession were recorded for the teeth and implants of patients from the mesiobuccal, distobuccal, mid-buccal, mesiopalatinal/lingual, mid-palatinal/lingual, and distolingual/palatinal regions. The plaque index was evaluated through visual inspection of the plaque accumulated in the gingival area, and the plaque was categorized into one of four grades. The gingival index was determined on the basis of color and tissue consistency, reflecting the severity of inflammation in the marginal gingiva. Bleeding on probing was assessed by gently inserting a UNC-15 periodontal probe (PCP15; Hu-Friedy, Chicago, IL, USA) into the gingival sulcus. All indices were measured during the examination. Calibration of the examiner was performed prior to the study to ensure reliability in probing depth and other measurements. The peri-implant and periodontal health statuses of the patients were examined. Healthy gingiva, which display an intact periodontium, exhibit minimal bleeding on probing (<10%) and shallow periodontal pocket depths (≤3 mm). In contrast, gingivitis is characterized by increased bleeding on probing (≥10%) with pocket depths

remaining ≤3 mm. Descriptions of periodontitis should encompass metrics such as the prevalence of bleeding on probing, the proportion of teeth with probing depths surpassing specified thresholds (commonly ≥4 mm and ≥6 mm), and teeth exhibiting CAL of ≥3 mm and ≥5 mm (*Papapanou et al., 2018*). Peri-implant health was characterized by the absence of erythema, bleeding on probing, swelling, and suppuration. The main clinical characteristic of peri-implant mucositis is bleeding on gentle probing. Erythema, swelling, and/or suppuration may also occur. As outlined in the 2017 World Workshop on Periodontology guidelines, in cases where prior examination data is unavailable, diagnosing peri-implantitis may rely on concurrent indications such as bleeding or suppuration during gentle probing, probing depths measuring 6 mm or greater, and bone resorption levels reaching 3 mm or beyond apically from the most coronal aspect of the intraosseous section of the implant (*Berglundh et al., 2018*). The implants were divided into three groups: peri-implantitis, peri-implant mucositis, and peri-implant health. Each group was evaluated according to periodontal status (periodontal health, gingivitis, and periodontitis). These parameters were chosen to test the hypotheses that peri-implantitis is correlated with periodontitis, that peri-implant mucositis is correlated with gingivitis, and that peri-implant health is associated with periodontal health. All the clinical examinations were conducted by a single clinician (T.Ş.) in a standardized manner to ensure consistency in data collection.

An assessment was conducted regarding the coexistence of periodontal and peri-implant conditions.

## Statistical analyses

Research analysis was conducted *via* the SPSS 26 (SPSS Inc., Chicago, IL, USA) statistical program. Shapiro-Wilk normality tests were performed to determine whether the data met the parametric test criteria. The study compared the three groups according to peri-implant health status. Paired sample t-tests were used to compare the implant and periodontal index values for each group. 95% CI values are shown in the table with the means and standard deviations. Cohen's d values were also calculated to evaluate the effect size. Chi-square test analysis was performed to compare peri-implant health samples according to periodontal status. The level of significance was set at $p < 0.05$. A total of 123 implants were analyzed, with 41 implants in each group on the basis of their peri-implant health status.

# RESULTS

## Demographic characteristic

The demographic characteristics of the study population are presented in Table 1.

## Distribution of groups

The implants were most commonly placed at #16 (12.2%), #36 (11.4%), or #46 (8.9%). The peri-implantitis, peri-implant mucositis, and peri-implant health groups comprised an equal number of patients (Fig. 1).

**Table 1 Demographic characteristics of participants.**

| | | f | % |
|---|---|---|---|
| Gender | Male | 39 | 57.1 |
| | Female | 30 | 42.9 |
| Education | Elementary school | 15 | 21.4 |
| | Middle school | 8 | 11.4 |
| | High school | 18 | 25.7 |
| | University | 28 | 40.0 |
| Work status | Working | 48 | 68.5 |
| | Nonworker | 5 | 7.1 |
| | Retired | 16 | 22.9 |

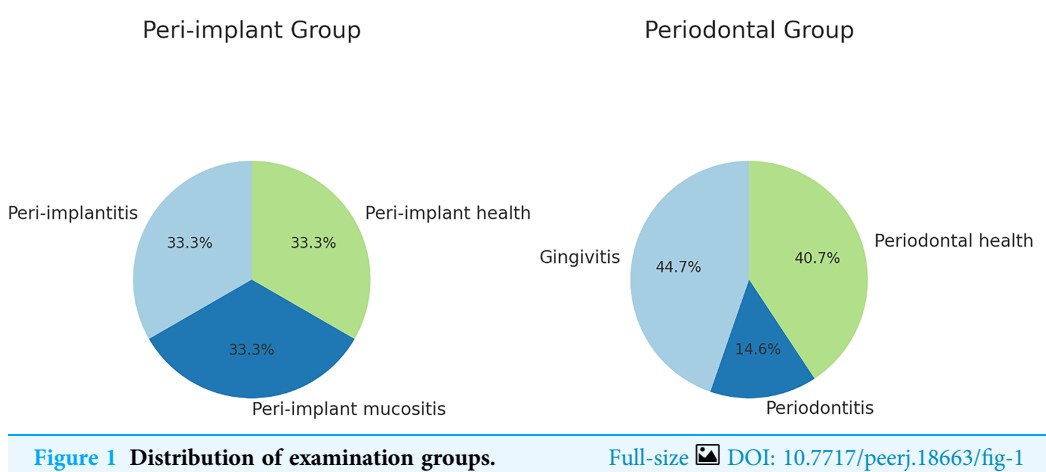

**Figure 1 Distribution of examination groups.**

## Comparison of periodontal and peri-implant conditions

Individuals with peri-implantitis had higher rates of gingivitis and periodontitis, respectively. Periodontal health was more common in patients with peri-implant health, whereas gingivitis was common in those with peri-implant mucositis. Among those diagnosed with peri-implant mucositis, the most common findings were gingivitis and periodontal health. A significant difference in the prevalence of gingivitis and periodontal health was observed between the peri-implant and periodontal groups, with peri-implantitis patients showing higher rates of gingivitis and periodontitis compared with the periodontal group ($p = 0.003$) (Fig. 2).

## Comparison of indices

### Plaque index

When peri-implant health and conditions were analyzed concurrently for plaque indices a significant difference was observed between the peri-implant and periodontal evaluations ($p = 0.001$), (95% CI for II [0.05–0.13], 95% CI for PI [0.19–0.30]). The plaque index during periodontal evaluation was greater than that during implant evaluation (Table 2).
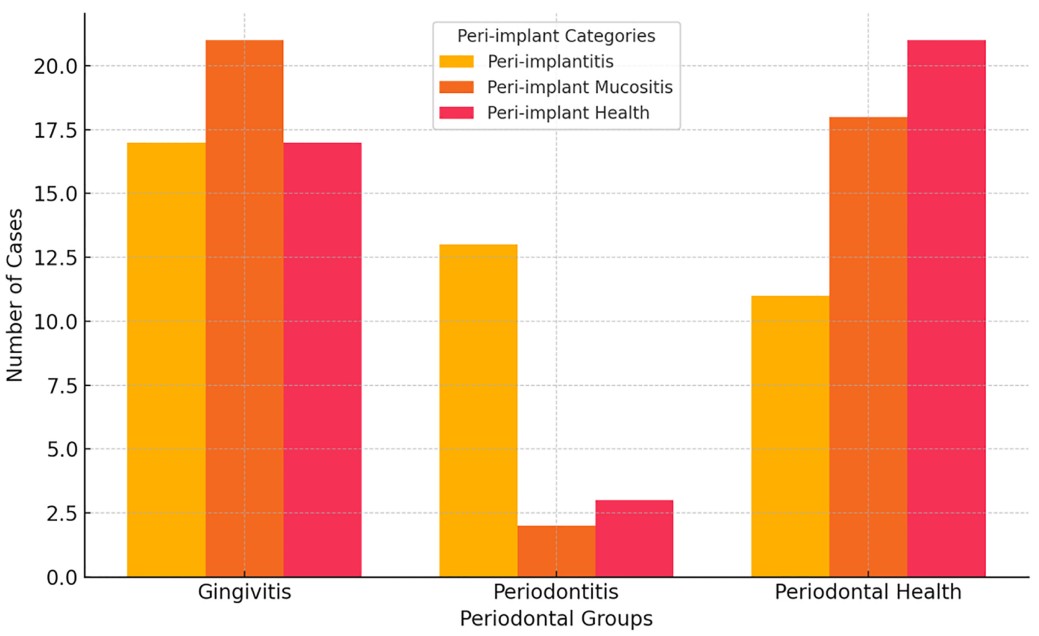

**Figure 2** Comparison of peri-implant and periodontal groups.

### Gingival index

A significant difference was observed in the context of gingival indices between peri-implant and periodontal evaluations in patients with peri-implantitis and peri-implant mucositis ($p = 0.001$), (95% CI for II [0.07–0.76], 95% CI for PI [0.07–0.14]). The gingival index during implant evaluation was greater than that during periodontal evaluation (Table 2).

### Probing depth

In patients diagnosed with peri-implantitis, a statistically significant difference in probing depth was observed between assessments of the peri-implant and periodontal regions ($p = 0.001$) (95% CI for II [3.24–4.59], 95% CI for PI [2.09–2.25]). The probing depth during implant evaluation was greater than that during periodontal assesment. There was no significant difference in probing depth between the implant and periodontal assessments in patients with peri-implant mucositis and peri-implant health ($p = 0.165$), ($p = 0.837$), (95% CI for II [2.00–2.28], 95% CI for PI [1.70–2.26]), (95% CI for II [1.82–2.12], 95% CI for PI [1.98–2.14]) (Table 2).

### Gingival recession

In the context of gingival recession, a significant difference was observed between peri-implant and periodontal recession in patients with peri-implantitis and peri-implant mucositis ($p = 0.001$), ($p = 0.014$), (95% CI for II [0.27–1.17], 95% CI for PI [0.06–0.31]), (95% CI for II [0.04–0.46], 95% CI for PI [0.05–0.18]). Periodontal evaluation revealed that gingival recession in implants was greater than that in teeth. Furthermore, no notable distinction was found regarding gingival recession between assessments of implants and

**Table 2 Demographic characteristics of participants.**

| | | İmplant index | | Periodontal index | | p |
|---|---|---|---|---|---|---|
| | | Mean ± S.D. | 95% CI (L-U) | Mean ± S.D. | 95% CI (L-U) | |
| Plaque index | Peri-implantitis | 0.13 ± 0.31 | [0.03–0.23] | 0.36 ± 0.48 | [0.21–0.52] | 0.001* |
| | Peri-implant mucositis | 0.09 ± 0.24 | [0.01–0.16] | 0.21 ± 0.22 | [0.15–0.28] | 0.002* |
| | Peri-implant health | 0.06 ± 0.14 | [0.02–0.10] | 0.16 ± 0.19 | [0.10–0.22] | 0.025* |
| | Total | 0.09 ± 0.24 | [0.05–0.13] | 0.24 ± 0.33 | [0.19–0.30] | 0.001* |
| Gingival index | Peri-implantitis | 0.42 ± 1.09 | [0.07–0.76] | 0.11 ± 0.11 | [0.07–0.14] | 0.001* |
| | Peri-implant mucositis | 0.26 ± 0.50 | [0.10–0.42] | 0.14 ± 0.17 | [0.08–0.19] | 0.023* |
| | Peri-implant health | 0.16 ± 0.42 | [0.02–0.29] | 0.09 ± 0.16 | [0.04–0.15] | 0.412 |
| | Total | 0.28 ± 0.74 | [0.15–0.41] | 0.11 ± 0.15 | [0.09–0.14] | 0.001* |
| Probing on depth | Peri-implantitis | 3.91 ± 2.14 | [3.24–4.59] | 2.17 ± 0.26 | [2.09–2.25] | 0.001* |
| | Peri-implant mucositis | 2.14 ± 0.44 | [2.00–2.28] | 1.98 ± 0.88 | [1.70–2.26] | 0.165 |
| | Peri-implant health | 1.97 ± 0.48 | [1.82–2.12] | 2.06 ± 0.25 | [1.98–2.14] | 0.837 |
| | Total | 2.67 ± 1.55 | [2.40–2.95] | 2.07 ± 0.55 | [1.97–2.17] | 0.006* |
| Bleeding on probing | Peri-implantitis | 0.66 ± 0.37 | [0.54–0.78] | 0.23 ± 0.25 | [0.15–0.31] | 0.001* |
| | Peri-implant mucositis | 0.62 ± 0.29 | [0.53–0.71] | 0.18 ± 0.15 | [0.13–0.23] | 0.001* |
| | Peri-implant health | 0.00 ± 0.02 | [0.00–0.01] | 0.14 ± 0.18 | [0.09–0.20] | 0.001* |
| | Total | 0.43 ± 0.40 | [0.36–0.50] | 0.18 ± 0.20 | [0.15–0.22] | 0.001* |
| Gingival recession | Peri-implantitis | 0.72 ± 1.41 | [0.27–1.17] | 0.19 ± 0.38 | [0.06–0.31] | 0.001* |
| | Peri-implant mucositis | 0.25 ± 0.66 | [0.04–0.46] | 0.12 ± 0.20 | [0.05–0.18] | 0.014* |
| | Peri-implant health | 0.16 ± 0.42 | [0.02–0.29] | 0.23 ± 0.47 | [0.08–0.38] | 0.410 |
| | Total | 0.38 ± 0.96 | [0.20–0.55] | 0.18 ± 0.37 | [0.11–0.24] | 0.001* |
| Clinical attachment level | Peri-implantitis | 3.97 ± 2.21 | [3.28–4.67] | 2.28 ± 0.36 | [2.16–2.39] | 0.001* |
| | Peri-implant mucositis | 2.09 ± 0.55 | [1.92–2.26] | 1.98 ± 1.02 | [1.66–2.30] | 0.869 |
| | Peri-implant health | 1.98 ± 0.61 | [1.79–2.17] | 2.02 ± 0.68 | [1.80–2.23] | 0.971 |
| | Total | 2.68 ± 1.63 | [2.39–2.97] | 2.09 ± 0.74 | [1.96–2.22] | 0.001* |

Note:
* 95% confidence interval (Lower bound-upper bound).

periodontal health in patients diagnosed with peri-implant health ($p > 0.05$), ($p = 0.410$), (95% CI for II [0.02–0.29], 95% CI for PI [0.08–0.38]) (Table 2).

### Clinical attachment loss

A significant difference was observed in clinical attachment loss (CAL) between peri-implant and periodontal evaluations in patients with peri-implantitis ($p = 0.001$), (95% CI for II [3.28–4.67], 95% CI for PI [2.16–2.39]). The CAL at the time of implant evaluation was greater than that at the time of periodontal evaluation. There was no notable contrast in attachment loss values identified between assessments of implants and periodontal conditions in patients diagnosed with either peri-implant mucositis or peri-implant health ($p = 0.869$), ($p = 0.971$), (95% CI for II [1.92–2.26], 95% CI for PI [1.66–2.30]), (95% CI for II [1.79–2.17], 95% CI for PI [1.80–2.23]) (Table 2).

***Comparison of gingival recession among the peri-implant and periodontal groups***

In the assessment of the implants, a notable distinction in gingival recession was observed among the peri-implant groups ($p = 0.016$). The peri-implantitis group presented the highest level of recession, which differed significantly from that of the other two groups, whereas the lowest gingival recession value was observed in the peri-implant health group.

Implant evaluation revealed a significant difference in gingival recession between the periodontal groups ($p = 0.020$). The periodontitis group had the highest gingival recession rate, which differed significantly from those of the other two groups, whereas the periodontal health group had the lowest gingival recession rate.

## DISCUSSION

This study tested whether peri-implant and periodontal conditions occurred simultaneously. Gingivitis was detected mainly in patients with peri-implant mucositis and peri-implant health. These results indicate that while most patients with periodontitis also have peri-implantitis, most patients with peri-implantitis do not have concomitant periodontitis. Investigations into peri-implant biofilms consider the importance of implant-related environmental factors, such as material composition, surface roughness, corrosion/tribocorrosion processes, and the potential for systemic migration of metal particles, all of which can influence biofilm formation, the peri-implant tissue response, and the long-term success of the implant. These factors play crucial roles in facilitating effective, implant-driven therapies for peri-implantitis, which are essential for mitigating the health burden associated with implant-related inflammatory conditions (*Kotsakis & Olmedo, 2021*). The architectural characteristics of dental implants differ from those of natural teeth, including differences in morphology, surface material, texture, and energy (*Robitaille et al., 2016*). Furthermore, dental implants differ from natural teeth in that they are decay-resistant, lack pulps that could serve as early pathology indicators or contribute to endodontic lesions, and lack a periodontal membrane (*Misch, 2014*). Periodontal tissues attach teeth to alveolar bone *via* the periodontal ligament and suprabony connective tissues, which include collagen fibers anchored to the root cementum. In contrast, osseointegrated dental implants lack these connective tissue attachments, with direct bone contact and no intervening connective tissues (*Klokkevold & Newman, 2000*). Compared with implants to natural teeth, implants are typically conical screws made of titanium and/ or ceramic, which are known for their increased surface roughness and decreased surface energy. Although roughness, energy, and composition are interrelated, each factor can independently influence bacterial colonization, gene expression, and community composition (*Larsson et al., 2022*). The presence of peri-implantitis in implants led to significantly elevated amounts of dissolved titanium in subgingival plaque compared to healthy implants. This finding indicates a strong association between titanium dissolution and peri-implantitis (*Kotsakis & Olmedo, 2021*). This titanium dissolution, when combined with factors such as mechanical stress, corrosion, and bacterial activity, may further exacerbate the inflammatory process, ultimately contributing to implant failure (*Chaturvedi, 2009*). Tribocorrosion and metal corrosion impact peri-implant biofilms,

potentially leading to peri-implant inflammation and implant failure through direct mechanisms (such as immune modulation) or indirect pathways (by disturbing the microbiome) (*Kotsakis & Olmedo, 2021*). Additional investigations are needed to elucidate the factors underlying titanium dissolution and the role of titanium corrosion byproducts in the progression of peri-implant inflammation (*Safioti et al., 2017*). Owing to limited clinical data, the incidence and development of peri-implantitis do not differ between modified and non-modified implant surfaces (*Schwarz et al., 2021*). Differences in the implant and natural tooth environments affect the simultaneous occurrence of periodontal and peri-implant diseases. In this study, these differences led to the occurrence of not only periodontitis but also gingivitis with peri-implantitis. In addition, contrary to the aforementioned studies and in support of the study hypothesis, patients with periodontal health in peri-implant health cases and patients with gingivitis in peri-implant mucositis cases were the most common findings.

*Meffert (1996)* reported that the bacterial flora linked to the implant and native tooth during illness are mostly identical and consist primarily of gram-negative pathogens, including *P. gingivalis*, *Porphyromonas intermedia*, and *A. actinomycetemcomitans*. This findings indicate that the subgingival microbiota compositions are quite comparable between the distinct periodontitis and peri-implantitis clinical groups. These similarities encompass potential "periodontopathogens", such as *Prevotella*, *Porphyromonas*, *Tannerella*, *Bacteroidetes* (G5), and *Treponema* spp. (*Yu et al., 2019*). In contrast, *Dutra et al. (2023)* observed a varied array of bacteria near infected implants, some of which were unculturable and previously unidentified. The presence of bacteria unrelated to periodontitis could instigate inflammation in peri-implant tissues, highlighting notable distinctions in the microbiota between periodontal and peri-implant regions. Additionally, a relatively high prevalence of opportunistic pathogens, such as Staphylococcus and Candida species, characterizes the microbiome associated with peri-implantitis (*Iuşan et al., 2022*). The structure of biofilms in peri-implantitis is more intricate than that in periodontitis. Although various bacterial species have been identified as potential pathogens in peri-implantitis, periodontopathogenic bacteria are less prevalent (*Koyanagi et al., 2013*). In periodontitis, bacteria from the red complex are vital pathogens, whereas they are not prevalent in peri-implant biofilms. There may be a confirmation bias in the dissemination of information regarding their presence (*Kotsakis & Olmedo, 2021*). Another study revealed no discernible difference in the occurrence of periodontal bacteria around implant sites in patients with peri-implant mucositis compared with patients with gingivitis (*Salvi et al., 2012*). Host-bacterial interactions shape unique microbiomes in both periodontal and peri-implant environments, indicating that differences in microbial composition are associated with health and disease, both individually and at the core microbiome level. However, diseases can facilitate the migration of periodontal bacteria into peri-implant sulci, or periodontitis can progress to peri-implantitis (*Robitaille et al., 2016*). As in the majority of studies, peri-implantitis was not found concurrently in the patient, which did not support the hypothesis of the study; however, periodontal health was observed in peri-implant health, and gingivitis was observed in peri-implant mucositis.

Various alterations in microbial populations influence the initiation and advancement of inflammatory reactions surrounding both natural teeth and dental implants. Furthermore, prior occurrences of periodontal disease exert an additional influence on modifying the immune reactions of peri-implant and periodontal tissues in response to the accumulation of biofilms (*Dutra et al., 2023*). The majority of the cells in both entities tend to be plasma cells and lymphocytes. However, neutrophil granulocytes and macrophages have been reported to be more abundant in patients with peri-implantitis than in those with periodontitis (*Berglundh, Zitzmann & Donati, 2011*). Implant plaque control effectively inhibits the formation of bacterial plaques on titanium abutments. The absence of inflammatory cell infiltrates in the peri-implant mucosa further highlights the ability of the junctional epithelium at titanium surfaces to form a barrier, preventing the formation of a subgingival infection in the absence of supragingival plaque (*Berglundh et al., 1991*). The mRNA levels of IL-6 and IL-1β are elevated in tissues affected by both periodontal disease and peri-implantitis. However, no significant difference in the expression of metalloproteinases or their inhibitors was detected among the studied groups (*Figueiredo et al., 2020*). Conversely, soft tissues around implants likely trigger more enhanced host immune responses, such as dominant macrophage infiltration, to promote osteoclastogenesis compared with those in periodontitis in another study (*Yuan et al., 2022*). In addition, IL-1 and TNF-α serve as sensitive indicators of bone loss adjacent to both natural teeth and dental implants (*Machtei, Oved-Peleg & Peled, 2006*). *Salvi et al. (2012)* similarly reported that IL-1β levels were the same in their study, while MMP-8 levels were greater around the peri-implant region. Although peri-implantitis and periodontitis share similarities in terms of clinical presentation and etiology, significant histopathological distinctions differentiate these two conditions (*Berglundh, Zitzmann & Donati, 2011*). These significant histopathological differences may affect the incidence of peri-implantitis and periodontitis at the same time. Histopathological studies do not support the hypothesis that peri-implantitis and periodontitis occur at the same time. The study results partly point in this direction.

Genetic variations in the Fmlp receptor (FPR1) gene are strongly linked to increased susceptibility to periodontitis and peri-implantitis (*Turkmen & Firatli, 2022*). The genetic variation within the IL-17A gene may influence the predisposition to peri-implant diseases (*Talib & Taha, 2024*). Additionally, alleles 1 and 2 of the IL-1A gene and alleles 1 and 2 of the IL-1B gene are statistically associated with the success or failure of dental implants (*Vaz et al., 2012*). Ten genetic polymorphisms of inflammation-related molecules, including proinflammatory cytokines and protease inhibitors, may have substantially influenced periodontitis. An individual may inherit several relatively common high-risk polymorphisms, resulting in a cumulative high-susceptibility profile for periodontitis (*Kinane & Hart, 2003*). To date, specific genetic variations consistently associated with periodontitis in certain populations include those within ANRIL, COX2, IL1, IL10, and DEFB1 genes. However, many proposed candidate genes for periodontitis lack robust validation or replication (*Loos et al., 2015*). According to a previous study, individuals carrying the G genotype exhibit increased susceptibility to periodontitis, whereas those

with the G/C genotype demonstrate a greater risk of peri-implantitis (*Turkmen & Firatli, 2022*). Genetic studies of peri-implantitis and periodontitis do not support this hypothesis. Due to the differentiation of the genetic profile in peri-implantitis and periodontitis, periodontitis may not be detected in every patient with peri-implantitis, as observed in this study. In addition, periodontitis is a risk factor for peri-implantitis.

Substantial evidence suggests an elevated risk of peri-implantitis among individuals with a previous history of chronic periodontitis, inadequate plaque control proficiency, or a lack of consistent post-implant therapy maintenance (*Schwarz et al., 2018*). Additionally, robust evidence indicates that periodontitis increases the probability of implant loss. Moreover, moderate evidence suggests that individuals affected by periodontitis exhibit elevated rates of implant-bone loss, thus establishing this condition as a predisposing factor for peri-implantitis (*Shiba et al., 2021*). Although the presence of periodontitis is a serious risk factor for peri-implantitis, this is not always the case, as found in this study.

Furthermore, it is imperative to consider immunological and histopathological distinctions when devising treatment strategies for peri-implantitis and periodontitis (*Berglundh, Zitzmann & Donati, 2011*). Following non-surgical interventions, the microbial makeup of periodontal and peri-implant sites is observed to undergo comparable alterations, transitioning from an abundance of periodontal pathogens to a composition akin to healthy sites (*Shiba et al., 2021*). Notably, in implants featuring rough surfaces, a previous history of periodontal disease negatively affects survival rates, despite the use of scaling and root planing procedures (*Young et al., 2021*). There have also been reports of disease progression or recurrence, as well as implant loss despite treatment (*Heitz-Mayfield & Mombelli, 2014*). Microbial, genetic, and immunological differences in peri-implantitis and periodontitis are reflected in the treatment of these diseases. The study's results confirm these differences, but they do not validate the study's hypothesis. Conversely, this hypothesis is supported primarily by patients with periodontal health among peri-implant health cases and patients with gingivitis among peri-implant mucositis cases.

This study has several limitations and strengths worth noting. A key limitation is the potential selection bias, as the sample was drawn from a single university clinic and may not represent the broader population. This bias could skew the results toward a more specific or narrower population, limiting the generalizability to a wider audience. The study population consisted solely of individuals from a university clinic, which may not represent the general population, particularly those from different socioeconomic or geographic backgrounds. Although random sampling and strict inclusion and exclusion criteria were employed, some bias likely remains, although its effect is probably small. Additionally, the study's cross-sectional design limits the ability to establish causal relationships between peri-implant and periodontal conditions. The limitation of this design is that it provides only a snapshot of the relationships at a single time point, making it impossible to infer long-term effects. Despite these limitations, the study has several important strengths. The use of well-established diagnostic criteria from the 2017 AAP/EFP World Workshop on Classification of Periodontal and Peri-implant Diseases and

Conditions ensures consistent and reliable assessments. Moreover, data collected by a single clinician, including various indices such as the plaque index, gingival index, probing depth, bleeding on probing, clinical attachment loss (CAL), and gingival recession, provide a comprehensive evaluation of the conditions. Importantly, this study contributes valuable insights into the co-occurrence of peri-implant and periodontal conditions. The finding that peri-implantitis is often absent in patients with periodontitis suggests a complex relationship between implant treatment and periodontal health, warranting further investigation.

## CONCLUSIONS

The higher incidence of gingivitis among patients with peri-implant mucositis and peri-implantitis in patients with periodontitis underscores the relationship between peri-implant and periodontal conditions. These findings indicate a close relationship between maintaining good periodontal health and peri-implant health, which could enhance the long-term stability of dental implants. A comprehensive approach, especially for patients with peri-implantitis, that addresses both conditions through preventive care and patient education is essential for achieving optimal outcomes. Further research exploring the distinct characteristics of peri-implantitis and periodontitis will help refine treatment strategies, paving the way for a more tailored and effective management of peri-implant diseases.

## ACKNOWLEDGEMENTS

The AJE and Editage Editing Service performed the English editing. This study was presented as a poster at The European Association for Osseointegration Congress on 24-26 October 2024.

### Funding

The author received no funding for this work.

### Competing Interests

The author declares that they have no competing interests.

### Author Contributions

- Tuğba Şahin conceived and designed the experiments, performed the experiments, analyzed the data, prepared figures and/or tables, authored or reviewed drafts of the article, and approved the final draft.

### Human Ethics

The following information was supplied relating to ethical approvals (*i.e.*, approving body and any reference numbers):

Bolu Abant İzzet Baysal University Clinical Researches Ethics Commitee.

## Data Availability

The raw data is available in the Supplemental File.

## Supplemental Information

Supplemental information for this article can be found online at http://dx.doi.org/10.7717/peerj.18663#supplemental-information.

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
