# Peer review of "Investigation of the relationships between peri-implant diseases, periodontal diseases, and conditions: a cross-sectional study"

_PeerJ, doi:10.7717/peerj.18663_

## Round 0.1 · original submission · Major Revisions

Please consider the reviewer's comments and revise accordingly.

Reviewer 1 ·

Basic reporting

Dear Author,
I have reviewed your manuscript titled "EXAMINING PERI-IMPLANT AND PERIODONTAL CONDITIONS CO-OCCUR: A CROSS-SECTIONAL STUDY" which you have submitted for consideration for publication. I suggest improving the title of the article to emphasize the objective. I recommend specifying a guideline such as STROBE in order to validate the study and facilitate future references. I suggest conducting a correlation between the type of prosthesis present in each evaluated implant, as this may justify some modifications in the bacteriological response for conclusion I recommend updating the references.

Experimental design

robust experimental design

Validity of the findings

valuable insights for the academic community

Additional comments

no comment

Reviewer 2 ·

Basic reporting

The abstract sentence "Results: While patients with peri-implant mucositis mostly had gingivitis and patients with peri-implant health had periodontal health, those with peri-implantitis mostly had gingivitis and relatively less periodontitis.", is slightly confuse, I recommend author paraphrase it for better clarity, an example: "Results: Patients with peri-implant mucositis predominantly had gingivitis, while those with peri-implant health exhibited periodontal health. In contrast, patients with peri-implantitis mostly had gingivitis, with a lower occurrence of periodontitis."

Experimental design

no comment

Validity of the findings

no comment

Additional comments

no comment

·

Basic reporting

Dear Authors,
Thank you for submitting your manuscript for consideration. I've reviewed your manuscript, and it presents an interesting study, which is a relevant topic. However, I do have some comments that you may need to address before it can be considered for publication.

- English language usage: The text has numerous grammatical errors and awkward phrasings. Professional editing is strongly recommended.

- Literature review: The introduction lacks focus on the specific research question. A more targeted literature review is needed to provide a clear rationale for the study.

- Article structure: While the overall structure is acceptable, the methods and results sections need reorganization for clarity.

- Tables are included but could be improved in presentation. No figures are present in the manuscript, which would be interesting.

- Although the manuscript presents relevant results, the connection to the initial hypotheses is not always clear. The study's objectives and hypotheses need to be more explicitly stated in the introduction to provide a clear framework for the results and discussion.

Experimental design

The study appears to be original primary research and falls within the scope of PeerJ's Life and Environmental Sciences section. However, consider the following:

- Research question: The specific objectives and hypotheses are not clearly stated in the introduction. This needs to be addressed to provide a clear framework for the study.

- Methods: More detail is needed, particularly regarding participant recruitment, inclusion/exclusion criteria implementation, and examination procedures.

- Statistical analysis: The description of statistical methods is brief and lacks justification for the chosen tests. More information is needed on how each analysis relates to the study objectives.

Validity of the findings

- Data robustness: While summary data are presented, information on raw data availability is missing. This needs to be addressed.

- Statistical concerns:
a) Justification for the choice of statistical tests is lacking.
b) There's no mention of corrections for multiple comparisons, which could inflate Type I errors.
c) Effect sizes are not reported, limiting the interpretation of the results' clinical significance.
d) Consideration of potential confounding factors through regression analysis is missing.

- Conclusions: The conclusion section does not effectively summarize the main findings or their implications. It includes speculative statements not fully supported by the presented data.

Additional comments

- There is a discrepancy between the study completion date registered on ClinicalTrials.gov (March 31, 2023) and the date reported in the manuscript (November 2023). This inconsistency needs to be explained and addressed.

- The abstract would benefit from a more structured format.

- Information on examiner calibration (if multiple examiners were involved) is missing.

- The discussion lacks a thorough exploration of the clinical implications of the findings.

- Consider adding appropriate figures to enhance the visual presentation of your data.

To improve the manuscript, I recommend:

- Clearly state the study's objectives, hypotheses, and the specific knowledge gap being addressed in the introduction.

- Expand the methods section, providing more detail on participant selection, examination procedures, and statistical analysis rationale.

- Revise the statistical analysis approach:
a) Justify the choice of statistical tests
b) Address multiple comparisons issue
c) Include effect sizes for significant results
d) Consider regression analysis to control for potential confounding factors

- Restructure the results section to clearly link findings to the study's objectives.

- Revise the discussion to more critically analyze the results in relation to existing literature and explore clinical implications.

- Rewrite the conclusion to accurately reflect the study's findings, avoiding speculation beyond what the data can support.

- Address the discrepancy in study completion dates.

- Thoroughly proofread and edit the manuscript to improve English language usage and clarity.

Addressing these issues will significantly improve the manuscript's clarity, scientific rigor, and adherence to journal standards. I believe these revisions are substantial enough to warrant a re-review of the manuscript once completed.

---

## Round 0.2 · accepted · Accept

Thank you for providing the modifications recommended by the reviewers.

·

Basic reporting

The manuscript now meets all basic reporting requirements with improved language, clear structure, well-presented figures/tables, and proper data reporting. References are comprehensive and current.

Experimental design

Research methodology is robust and clearly described. Sample size calculation, inclusion/exclusion criteria, and measurement procedures are well-detailed. The study objectives and hypotheses are now explicitly stated.

Validity of the findings

Statistical analyses are appropriate with proper reporting of effect sizes and confidence intervals. Results are presented clearly with supporting figures, and conclusions align well with the findings. Study limitations are properly acknowledged.

Additional comments

No comments.